# Ethnic Background of the Two Feeding Stories in Mark's Gospel

Paula Andrea García Arenas

Faculty of Theology, Pontifical Xavierian University, Bogotá 110231, Colombia; garcia.paula@javeriana.edu.co

**Abstract:** The analysis delves into the conflict inherent within the thematic discourse surrounding the two tables as portrayed in Mark's Gospel, with particular emphasis on the section concerning the multiplication of loaves of bread (Mk 6–8). Noteworthy is the conflict arising from the juxtaposition of Jewish and pagan individuals at a shared table. This theological tension finds resonance in the narratives presented by Paul in Galatians and Romans, albeit Galatians 2:9 intimates a seemingly facile resolution, a departure from the intricate portrayal in Mark's Gospel. Mark's narrative accentuates two salient dimensions: firstly, the ethnic substrate of the conflict, and secondly, its contextual specificity within the historical milieu of Syria after the Jewish war. The ethnic genesis of this conflict, as delineated in the accounts of Flavius Josephus, furnishes a background essential for comprehending the dual incidents of bread multiplication: the initial instance catering exclusively to Jews and the subsequent occurrence inclusive of both Jews and other disparate ethnic groups "from afar" (Mk 8:3). The spatial symbolism in the section pertaining to the multiplication of loaves may symbolically represent the heterogeneous composition of the recipients, thereby exacerbating the challenges inherent in reconciling conflicts rooted in ethnic diversity.

**Keywords:** Paul; Mark's Gospel; ethnic background; ethnic diversity; conflict; loaves of bread; Jewish war; Flavius Josephus





## 1. Introduction

This research follows on from a previous one on "the historical background of the loaves section in the Gospel of Mark" (García 2016), which concluded that, according to the above, it would be important to find out for what purpose the evangelist has composed it. If the macro-narrative of the Gospel is considered, it is evident that Mark aims to lead readers, both real and implied, to the discovery of the identity of Jesus, the Messiah, the Son of God (Mk 1:1). In this section, Mark proposes a narrative element that gives unity to the section: the bread. According to Santiago Guijarro, "it is not about the material bread at all, but about something else, and so the disciples—and the readers—are invited to understand" (Guijarro 2012, p. 242). Thus, Mark, in the section on the loaves, has woven two stories together to achieve the goal of revealing the identity of Jesus, and at the same time, these cycles are concerned with arriving at a specific understanding of that identity, but from two different propositions. In this research, after presenting the debate on the geographical location of the addressees of Mark's Gospel, the Syro-Palestinian region is taken as a reference point, and the development of the Gospel is located there.

In this research, it is also established as a precedent that the chronological placement of the various books of the New Testament is understood as follows: the Letter to the Galatians, the Letter to the Romans, the Gospel of Mark, the Book of Acts of the Apostles, and the Letter to the Hebrews. This alignment is made with reference to the Jewish war exposed in the writings of Flavius Josephus, aligning with the time of the Gospel of Mark. However, in order to present an overview of the coexistence of nascent Christian groups of different origins throughout the first century, reference will be made to these New Testament writings.

### 2. Two Feeding Stories in Mark's Gospel

To discern the issues faced by the addressees of Mark, delineate their historical context, and elucidate Mark's persuasive intent, the inquiry begins with an interrogation: why does Mark incorporate two accounts of the feeding of the multitude into his narrative, and what causes the discordance between them? To address this query, a presentation will be made, grounded in literary analysis, elucidating the distinctive characteristics of each feeding episode within Mark's composition. Subsequently, a literary analysis will be provided, outlining the tripartite structure of the bread narrative spanning Chapters 6, 7, and 8, with particular emphasis on Chapter 7 in juxtaposition to Chapters 6 and 8 (Wheatley 2023). In addition, the meaning of the epithet "Syrian" attributed to the woman who appears in Chapter 7 will be explored, guided by the indications provided by Flavius Josephus on the divisions and disagreements in the Jewish population of Galilee. The tripartite structure of the section on the loaves (Mk 6:30–8:26) is as follows:

- First story of the multiplication of loaves (Chapter 6);
- The pure and the impure and the Syrophoenician woman (Chapter 7);
- The second story of the multiplication of loaves (Chapter 8).

  With the following duplicates:

- Two stories of the multiplication of loaves (Mk 6:30–44; 8:1–10);
- Two stories of crossing the lake (Mk 6:45–56; 8:10);
- Two stories of controversy with Pharisees (Mk 7:1–23; 8:11–12);
- Two stories dealing with bread or leaven (Mk 7:24–30; 8:13–21);
- Two healing stories (Mk 7:31–37; 8:22–26).

Fowler (1981) speaks of "duplicates" and "cycles". For him, duplicates are "two variants of a single story", while a cycle is a story that includes several events and returns and repeats itself at other times and places within the gospel. It is emphasized that while duplications are recurrent throughout the Gospel, the conspicuous resemblance in both form and content between the two narratives detailing the multiplication of loaves is notable:

> When one assumes the two-source hypothesis, one concludes that Matthew and Luke borrow from the story found in Mark. Matthew incorporates the two accounts: the feeding of the five thousand (Mt 14:13–21) and the feeding of the four thousand (15:32–39), as well as all the material of the section. Luke, on the other hand, refers only to the feeding of the five thousand and departs from Mark's storyline immediately after the first feeding story. He then resumes his use of Mark only with the episode at Caesarea Philippi, thus omitting Mk 6:45–8:26. (Fowler 1981, pp. 5–6)

This is in addition to the fact that the Johannine version seems to present an account that includes both of Mark's stories. Iverson, for his part, identifies within the section some characteristics referring to the Gentiles: the Syrophoenician woman (Mk 7:24–30), the deaf man (Mk 7:31–37), and the 4000 of the second multiplication (Mk 8:1–9) (Iverson 2007, p. 152). In addition to Iverson's intuition, Fowler argues that the theories proposing cycles behind the so-called "loaves section" and their obvious differences fail to resolve the problem of the connections between these two feeding stories. Therefore, it cannot be concluded that the pre-eminent duplicate feeding stories are variants of traditional stories. Rather, one story is traditional (Mk 8:1–10), and the other is a composition by Mark (Mk 6:30–44). The evangelist has composed his own story as a backdrop for the traditional story, thus controlling the way the reader perceives it (Fowler 1981, p. 181; García 2014, pp. 321–23).

According to the above, in the case of the loaves section, the narrator's intention to lead his readers (real and implied) to the discovery of the identity of Jesus, the Messiah Son of God, is identified, but under a narrative element that gives unity to the section: the bread. According to Guijarro, "it is not about the material bread at all, but about something else, and so the disciples (and the readers) are invited to understand" (Guijarro 2012, p. 242).

Thus, Mark, in the section on the loaves, has interwoven two stories to achieve the goal of revealing the identity of Jesus, and at the same time, these cycles are concerned with coming to a specific understanding of that identity: The Messiah, Son of God (Mk 1:1) (Focant 1992, pp. 1039–63; Fowler 1981, p. 5).

As for the place, the one that frames the section in Mk 6:31–32 is "a desert place" indicated by the lake, which is not necessarily a desert (as in Mk 8:4) but is uninhabited. This withdrawal of Jesus into solitude is something the narrator has performed before (Mk 1:12.13.35.45), making it something "habitual" of Jesus for the reader. The crowd and the boat are thematic points of interest for the evangelist (Mk 2:2; 3:7f.20; 4:1f.). What is striking is the passive role given, at the level of the narrative, to the multitude at the first multiplication, for the account only says: "they all ate their fill" (Mk 6:42). The commentary after the second multiplication is almost identical (Mk 8:8). There was no reaction. Not even the disciples commented. This puts us on a plane of meaning, narratively speaking, beyond the miraculous action performed by Jesus (Yarbro Collins 2007, p. 319; Pérez Fernández 2008, p. 410; Focant 2004, p. 225).

As indicated above, the tripartite structure of this section places Chapter 7 right in the middle, where the issue of what is "pure" and what is "impure"—referring to food and people—is addressed. Furthermore, in this section, it can be observed that two groups are fed with the bread: those who are "like sheep without a shepherd" (Mk 6:34) and those "who have come from afar" (Mk 8:3). However, particular attention will be directed towards the entreaty voiced by a Syrophoenician woman for the remnants of the bread, aimed at discerning indications of receptivity or resistance towards Gentiles within the community to which the Gospel is directed. This analysis seeks to identify the interpretive frameworks provided by the narrative in Chapter 7, facilitating a deeper comprehension of the section delineated by Mark.

The episode begins with an indication that reflects the characteristic concern of Mark that Jesus should go unnoticed (Mk 1:35; 6:32.46; 8:13). It is pointed out in a very general way that Jesus, to go unnoticed, enters a house but cannot remain hidden (Mk 1:45; 5:43; 9:30). The evangelist is not trying to show here that Jesus transgresses the Jewish prescriptions of purity because the house was pagan, but that his intention runs parallel to the messianic secret and touches on the idea of revelation. The same was said in Mk 1:45–2:2 in relation to the Galilean sphere. Gnilka (1992) and Yarbro Collins (2007) assert that verse 24a, which situates the narrative in Tyre, is purposeful on Mark's behalf. This assertion is supported by the fact that in Mark 3:8, individuals approached Jesus from the vicinity of Tyre, whereas in the present context, Jesus travels there alone (Gnilka 1992, p. 338; Yarbro Collins 2007, p. 364). The inhabitants of Tyre were Phoenicians, and they held the worst reputation among the Jews (Is 23; Joel 4:4–6; Zech 9:2), although with some promises of salvation (Psalms 87:4).

Continuing the narrative trajectory prompts the following inquiry: why does Jesus journey to the region of Tyre? Alonso, in his study of this episode, posits three potential explanations. Firstly, it could be inferred that Jesus seeks solitude and tranquility. This inference is supported by the account in 6:31, wherein it is stated that he desires a moment of respite with his disciples, even seeking time for sustenance. This required tranquility is interrupted in the narrative by various crowds: that of Mk 6:33; 6:54; 7:1.14. Furthermore, this need for quietness manifested by Jesus is reinforced by the narrator in Mk 7:24 when he indicates that Jesus wants to go there and go unnoticed. According to this, it could be thought that Jesus in pagan territory would not be recognized (Alonso 2011, p. 313) or that the displacement to pagan territory on the part of Jesus in Mk 7:24 can be attributed to his own desire to avoid being recognized.

Another response offered by Alonso is the desire to withdraw and reflect. This is also likely in the narrative, given that the action begun in Mk 6:7–13, which was taken up in Mk 6:30, has not been completed by Jesus. They have not had Jesus' feedback on what they told him. Jesus has not had the opportunity to instruct them. According to the discerned rhetorical intentionality within Mark's narrative, it is plausible to consider that this entire

section, and indeed the entirety of the Gospel, may already serve as instructional material. However, further exploration of this proposition will be deferred to a later discussion.

A final response offered by Alonso is precisely the formation of his disciples. Suppose the narrative trajectory is followed, wherein Jesus, subsequent to delivering a discourse on the supersession of Jewish purity regulations pertaining not solely to dietary practices but also to individuals deemed impure, it becomes evident that this is not merely a reflective observation. Rather, it may be part of Jesus' pedagogical strategy to take them to pagan territory to show them that this proposal can be fulfilled and that he himself is willing to do so.

Whatever the answer, it is clear that the episode of the Syrophoenician woman will be decisive, not only for understanding the meaning of the section on the loaves but also for enabling the comparison between the bread and the healing of the woman's daughter. For the claim of the Gentile world represented as "*puppies*" (κυνάριον) have the right to bread and, above all, for the narrative opposition that this woman will make in her response as the one who understands the collectivity of discipleship that the disciples themselves seem not to have understood. The inquiry regarding whether the term "*puppies*" references the epithet Jews employed for Gentiles as "*dogs*" (1 Samuel 17:43; Job 30:1; 2 Kings 8:13) cannot be definitively addressed, as there exist instances where legal scholars applied this designation to the lay populace lacking knowledge of the law. It is apparent that a comparison is being made here rather than an allegory (Gnilka 1992, p. 341).

What Jesus says in Mk 7:27, "let the children first be filled", may mislead and even confuse the reader, for it takes away the severity of the following image by making a concession and transforming the refusal into a hierarchy. However, Jesus' initial rejection of the woman is less categorical than in the subsequent symbolism of the bread. In Mark's intervention, the necessary historical-salvific ordering of the pagans after the Jews is affirmed. It does not affirm their rejection. Picking up a widespread self-classification of the Jews as children or sons of God, he speaks of them as sons entitled to be the first to be filled. The historical salvific anteposition of the Jews is in complete agreement with the Apostle Paul, who calls the Gospel the power of God for the salvation of everyone who believes "for the Jew first and also for the Greeks" (Romans 1:16; 2:9).

Mark presents the woman as Greek and Syrophoenician. Neither of these two qualifications fits well as an ethnic designation. Matthew (15:22) met the difficulty by presenting her as a Canaanite woman from that region. It would be more appropriate to take the word "Greek" as an indication of her culture and religion. She was a Hellenized native and belonged to the higher social stratum. The reader is left in no doubt that she was a pagan, not a Jewess. But this pagan woman, like the hemorrhagic woman, throws herself at the feet of Jesus (5:33) (Gnilka 1992).

Rhoads says that the woman responds to Jesus' riddle to show that she has understood the question of the loaves (Rhoads 1976). Alonso points out the woman's ability to listen (Mk 7:25) and respond (Mk 7:28–29) display attitudes that contrast with that of the disciples who listen but do not understand (Mk 7:14,18; 8:18) and with that of the Pharisees who "refuse to eat the bread without having done the ritual purification while she is happy with the crumbs" (García 2014, p. 317). The foreign woman dares to ask for what is proper for the locals (Jews): Salvation. The "it is not right" pronounced by Jesus, according to his own cultural parameters, is surprising, but he understands that this foreigner also has the right to salvation (García 2022, p. 15). Estévez will say that the women, in a special way, embody the characteristic virtues of Jesus' disciples, exemplified in the Gospel healings, by offering the essential guidelines for the behavior of true believers (Smith and Choi 2020). This is evidenced by Mark's account of the Syrophoenician woman, portrayed as worthless through the image of the "*dogs*" but presented by Jesus by her great faith and humility as a model to imitate, even above the disciples and the Gentiles themselves (García 2020, p. 146).

However, not only the identity of Jesus but also his deep connection with the salvation He offers—evident in the plea of the Syrophoenician woman—and the universal character

of this salvation is extended to both Jews (*sons*) and non-Jews (*little dogs*). While many scholars attribute this to a possible reluctance to embrace the Gentile world, it is posited here that underlying this issue, there may be an ethnic differentiation (Brett 1997; Hockey and Horrell 2019). The Jewish-Roman conflict, which engendered animosity between individuals of divergent ethnic backgrounds, likely compounded the challenge of fostering such inclusivity during the first century C.E. Now, attention will be directed towards some examples of texts in which ethnic backgrounds presented a problem or how "two sides" appear to be identified.

Thus, Mark's ethnic designation of the woman in Chapter 7 as Greek by birth and Syrophoenician in order to differentiate her from the "sons" (Jews) suggests that in the context of Mark's Gospel, the background of the new believers was apparently an issue and that Christians coming from Judaism would have had some kind of privilege and "suspicion"—not good—towards Christians coming from the Gentiles. In the subsequent section, the presentation of this ethnic issue within its context will be examined.

### 3. "Ethnic" Problem in the Mark's Gospel Context

The social sciences understand conflict with other groups as something that contributes to establishing and reaffirming the identity of the group itself, maintaining its boundaries in relation to the surrounding social world (Coser 1956, p. 23). This emphasis on the "construction" of identity—race, culture, nationhood, religion, etc.—including that of "ethnicity", has taken hold in scholarly discourse. The idea of "ethnicity" has been seen to be particularly apt both for ancient Judaism and as a framework for exploring its offshoot, Christianity (Lieu 2002, p. 4; Hockey and Horrell 2019). Although concepts such as "ethnicity" and "identity" are not the same in the ancient world, their application has provided benefits. In contrast to older accounts, which presented Christianity as "neither Jew nor Greek" because it is problematic to be restricted to any ethnic group, the use of rhetorical strategies has meant that Christian texts can be treated as equally susceptible to ethnic or identity-based analysis. For example, the particular use in such early Christian writings of the language of "people" or of "race", as well as that of "family", has received particular attention. It is the group identity that is of primary interest here rather than the self-awareness of the individual, even if the two are recognized to intersect with each other (Lieu 2002).

Jewishness became an ethnoreligious identity and may be "said to cohere on the basis of common culture and collective will". Cohen speaks of six characteristic features of ethnic groups: (1) "a named group"; (2) "attached to a specific territory"; (3) "a sense of common origins" or common ancestry; (4) "a common and distinctive history and destiny"; (5) "one or more distinctive characteristics"; (6) "a sense of collective uniqueness and solidarity". Cohen (1979) assumes that these six features characterized ancient Ἰουδαῖοι and can be mapped onto the Greek term ἔθνος (*ethnos*) without further ado as if the semantic ranges of ethnos and "ethnicity" are identical. Although even externally verifiable cultural differences are "imagined" because the community must determine which characteristics form the boundary between it and other groups, Cohen also assumes that the six features are common to all ethnic groups. The most important feature, however, is a belief in common ancestry because it alone is unchangeable (Miller 2014, p. 221).

In this sense, to speak of a "Jew" is not a simple matter, as it does not only refer to the individual who was born in a certain place or professes a certain religion. It will have to do with the reference group (Frey et al. 2007). The ambiguity lies between its reference to those who came from the region around Jerusalem and those who came from the whole of Palestine, as became the case in the days of Hasmonean, Herodian, and Roman rule. Some refer to the term to a place, which is much less confusing than to religion. Such is the case with Josephus, who uses the "Judaic" term Ἰουδαῖος to denote a person by reference to his geographical origin. In this sense, Schwartz's words are accurate (Schwartz 2007, pp. 10–11):

Note A.J. 18.196, where someone learns that Agrippa was "an *Ioudaios* by *genos* and one of the most distinguished people in that (land)" . . . As Cohen noted, if this sentence is to make any sense, *Ioudaios* must mean 'Judaean'.

However, such passages that clearly take *Ioudaios* to mean Judaean are few and scattered in Josephus' later writings. In contrast, there is more evidence for *Ioudaios*, meaning Judaean, in the context of his writings about the Roman-Jewish war of 66–73/74CE, which culminated with the destruction of the Second Temple and ended with the fall of Masada. Josephus clearly calls this war the *polemos Ioudaikos* (A.J. 20.258, Vita 27, 412–413; cf. A.J. 13.173), and this plainly means, in our terminology, the Judaean War.

However, Miller points to some sympathetic repair, which is critical:

It is a mistake to combine the translation question with the more important and logically prior question of the meaning of *Ioudaios* in the Greco-Roman world. If we assume, for the sake of argument, that ancient readers were aware of distinct meanings of *Ioudaios*, which correspond exactly to our English terms "Jew" and "Judaean", we must remember that there was no simple way of conveying this distinction in Greek". (Miller 2010, p. 99)

In the Galilee case, Josephus does not identify the inhabitants of the main cities of Galilee, such as Sepphoris, Tiberias, and Gabara, as Γαλιλαιων. In fact, the inhabitants of the main civic centers, under recurrent threat from the Galilean population, are frequently contrasted with the Γαλιλαιων. Josephus saw that the inhabitants of Sepphoris, who were Galileans, decided to seize that territory as booty because of their alliance with the Romans because they were engaged in a "struggle for their native place". The Γαλιλαιων were, apparently, country dwellers, docile villagers full of resentment towards those of Tiberias and Sepphoris. Nevertheless, from a geographical perspective, the inhabitants of Sepphoris, Tiberias, and Gabara were just as much Galileans as their rural neighbors (Thiel 2020, p. 221).

In a general way, Josephus speaks of "Judaeans" to refer "externally" to one people as opposed to another, but when speaking of the same people "internally", he prefers to speak of "Jews". Indeed, when references to pagans in Judea arise, the terms "Greeks" and "Hellenes" are notably employed, thereby corroborating the assertion (Frey et al. 2007, p. 22).

This leads to the conclusion that in the New Testament overall and within Mark's Gospel specifically, when designations such as "Jews" or the more specifically religious term "Pharisees" are encountered, they are juxtaposed with "non-Jews" in a general sense, without necessitating specification of their origin in a particular manner. Such is the case with the expression in Mk 7:29, where "*puppies*" include, in a general way, those who do not come from Judaism.

It can be posited that the conflict underpinning Mark's Gospel, delineated narratively through two distinct groups—one comprising Jews and the other non-Jews—likely stems from a deeper underlying context. Is this context defined by issues of ethnic origin? Or is it shaped by the Jewish-Roman conflict that transpired between 66–67 CE, culminating in the destruction of the Jewish Temple in 70 CE? These two inquiries will be explored subsequently.

### 3.1. Is the Table Problem an Ethnic Problem?

As previously noted, within the New Testament, two major opposing groups are discernible: Jewish Christians or Christians coming from Judaism and pagan Christians or Christians coming from the Gentiles, often called "Hellenists" (Acts 6:1–2).

At this point, it should be noted that while many Christians coming from Judaism found it difficult to integrate with the Gentiles, others did, in fact, do so. Barclay speaks of three scales to refer to the degree of Hellenization of the Jews. These scales measure assimilation, acculturation, and adaptation. Assimilation refers to social integration and

the degree to which Jews abandoned their social life, which was limited to the Jewish community. Acculturation refers to language and education, assessing proficiency in Greek and literature, etc. Accommodation determines the level of use of acculturation, ranging from immersion to antagonism towards Greco-Roman culture (Barclay 1996, p. 127).

Romans 14:1–18 speaks of the context of meals in Rome. And, even in a much later writing, which probably dates from the end of the first century, this problem is still presented, as in the case of Heb 13:9b: "For it is well for the heart to be strengthened by grace, not by regulations about food, which have not benefitted those who observe them". What is most striking is that the verb in v.9a is in the present tense, suggesting that this is still relevant to the contemporary situation. Furthermore, "grace" is contrasted with "food", and the heart is spoken of as the interior of man as in Mark (Mk 7:6.19.21).

Another text that serves as a contextual reference is Gal 2:11–14, where Peter, it seems, is in Antioch sharing a table with Gentiles when a group of conservative Jewish emissaries, called "those of James", arrive from Jerusalem. The episode in Gal 2:12a clearly takes place in the context of eating and sharing a table with pagans (Núñez Regodón 2002, p. 105). According to Schenke, it is very unlikely that all Judeo-Christians accepted the practice of eating with pagans, and he suggests that even before Gal 2:11–14 there were meals alone for those Judeo-Christians who refused to share the table with pagan Christians (Schenke 1999, pp. 494–95). This leaves us with the picture of at least two "table groups". Could this be the case within the Marcan community? This consideration would undoubtedly aid in elucidating the rationale behind the presence of duplicates or doublets within his gospel. While it has been observed that these duplicated episodes serve a distinct redactional and narrative purpose centered around the instruction of the Twelve, it is also evident that they harbor a discernible rhetorical intention aimed at the actual recipients or readers of his gospel, intended to persuade them of a particular notion. Is Mark seeking to persuade them that there exists only one shared communal table?

As to whether the incompatibility of table fellowship over meals was so great as to differentiate between two groups, Dunn, referring to the Antioch community, offers three possibilities (Dunn 1983, pp. 10–18).

First, it is suggested that the common meals between Jewish and Gentile Christians completely transgressed all the laws of table fellowship, including the Levitical or "greater" laws, from the possibility of eating pork to eating meat sacrificed to idols to eating meat from animals improperly slaughtered according to Jewish law. At the opposite extreme, the possibility is that the table fellowship at Antioch involved strict observance of the dietary laws, even the halachic prescriptions concerning ritual purity, in which case James' demands reported in Gal 2:11–14 and Acts 15 would go beyond dietary matters to include circumcision. The middle option, which, for Dunn (1983), is the most likely, assumes that the meals did not transgress the laws so openly, but neither were they strictly observed either. In that case, the Jerusalemites would have demanded a more rigorous observance, especially about ritual purity. This seems to be the case in Mk 7:1–23.

For Schenke (1999), Paul testifies in Gal 2:11f that the Judeo-Christians shared the table with Christians coming from paganism, which implied postponing some exclusive ritual matters in favor of the communion of all believers. This account also shows that Peter was always willing to participate in the common meals of the church in Antioch, together with Gentile Christians, i.e., following a practice that was already taking place in that church when he arrived there, but evidently, this was not enough to counteract the pressure exerted on him by these Christians in Jerusalem (Dunn 1983; Schenke 1999).

The questions that arise from this picture are as follows: did all the believers in Antioch act in this way? And did the question of how they could live together regarding ritual norms arise before the appearance of the people of James? Schenke (1999) takes the latter for granted. On the former, he says that there is no reason to assume the general validity of a common table between Jews and pagan Christians. He continues as follows:

> There may have been, along with mixed dining, meetings, and celebrations reserved for Judeo-Christians (...) The common agape was held in private homes

and, therefore, in small groups, rarely exceeding twenty or thirty persons. It is not likely that in all these small groups, Judeo-Christian and pagan Christians came together. (Schenke 1999, p. 495)

This answer on a theological, but possibly not historical, level is provided by Luke in the book of Acts, where, from a vision in which God himself speaks, Peter is warned, among other elements, that God's will be the common table.

In Luke's presentation of the beginnings of the Christian communities, it is, of course, quite natural that this question of relations with the Gentiles should arise in a cosmopolitan city like Antioch, and it would seem intrinsically plausible that it arose there rather than in Jerusalem. By presenting in Jerusalem a figure like Nicholas, a proselyte of Antioch and one of the Seven (Acts 6:5), one can easily see how the issues discussed in the church at Antioch could have been made known in Jerusalem. However, if the issue had been raised in Antioch and then found its echo in Greek-speaking Judeo-Christian circles in Jerusalem, then it is to be expected that any relaxation of the rules applicable to Jewish contacts with Gentiles would have come at the height of resistance in Jerusalem (Marcus 2004, p. 68).

From the above, with the entry of the pagans into God's new community, the whole law was affected, especially the ritual precepts that drew a clear boundary between Jews and pagans. When Judeo-Christians and pagan Christians lived together in a community and shared the same table, the problem of the food precepts that divided the two groups had to be resolved. The legal distinction between pure and impure meals practically prevented a Jew from eating with a pagan. In this situation, Schenke (1999) suggests two possible solutions.

The first was that the pagan Christian should also observe the ritual rules on food. This was the solution offered in the so-called "Apostolic Decree" of Acts 15:28f, although it is not known from when and in what sphere it was in force. Secondly, the legal distinction between pure and impure food was not applied to the pagan Christians. It seems that the Antiochene community defended this conception and that Peter temporarily shared it, as presented in Gal 2:11 (Schenke 1999, p. 290).

Thus, Acts 10:9–16 is a redactional attempt to reconcile this problematic commensality, as is the halachic discourse of Mk 7:14–23. Both accounts owe their present form to their authors. In Mark, especially the passage from public discourse to private instruction (Mk 7:14.17), the formula of attention Mk 7:16 (Mk 4:9), the rebuke to the disciples Mk 7:18f (Mk 4:13), and the commentary of the reporter (Mk 7:19c); in Acts, all the preparation from Acts 8,1 to the statement in Acts 11:20: "even to the Greeks (Hellenists)".

Both accounts (Mk 7:1–23 and Acts 10:9–16) are, in any case, apparently insufficient to further the question of the admission of the Gentiles to the church since the terms of their admission are condensed in Acts 15, despite the retelling of the story in 11:1–18 and the apparent agreement of the Jerusalem church, where God had, in fact, acted to convert the Gentiles (Gal 2:2–10; Acts 11:18).

However, it seems that the admission of the Gentiles was not the problem for these communities, for Jesus had accepted and recognized the faith of a pagan of Phoenician origin (Mk 7:27), and Peter had accepted the hospitality of a Gentile (Acts 11:3). The problem they seem to face is what to do with these people after they have been admitted. Thus, it is understandable that it is Act Chapter 15 that deals with this question of whether Gentiles could be admitted to the church without circumcision and obedience to the law (15:1.5) but concludes with an agreement—the "Apostolic Decree"— which says nothing explicitly about circumcision or obedience to the law, but deals with the rules and regulations that would allow Jewish and Gentile Christians to live together and to eat together (Wedderburn 1993, p. 374).

In short, if in what is presented by the book of Acts, Peter was urging eating with Gentiles, perhaps the original meaning of the story was not so decisive since, judging from what is presented in Gal 2:11–14, Peter was not sufficiently convinced on the issue and, judging from the acts depicted there, the Jerusalem church was not sufficiently convinced

on the issue of the admission of Gentiles to the church because of the consequences this brought, not for salvation, but in practical terms: at the table.

But is this "difficulty" of accepting Gentiles at the same table only an ethnic connotation, or is there something else behind this "difficulty"? It will be examined below whether the aftermath of the Jewish warfare has influenced the ethnic conflict that has been identified.

*3.2. Is the Table Problem Influenced by the Jewish War (66–67 C.E.)?*

The above picture, while exposing the obvious ethnic separation in the New Testament texts, is best explained using the contributions of the Jewish historian Flavius Josephus, with the due recommendation to read his "narrative" with some discretion, since as a historian he followed the usual Greek practice, namely that "he was expected to vary the diction of his source, to embellish the narrative, to create something new" (Cohen 1979, p. 233).

At this juncture, it is necessary to depart from other New Testament texts and focus attention on the Jewish and non-Jewish factions underlying the target community of Mark's Gospel, thus reflecting the background of the two feeding narratives introduced within the Gospel account. The inquiry will revolve around whether the context is influenced by the Jewish war, especially if Mark's community is situated in Syria, located north of Palestine. The writings of Flavius Josephus shed some light:

> Another disturbance occurred at Cesarea, where the Jewish portion of the population rose against the Syrian inhabitants. They claimed that the city was theirs on the ground that its founder, King Herod, was a Jew. Their opponents admitted the Jewish origin of its second founder but maintained that the city itself belonged to the Greeks since Herod would never have erected the statues and temples that he placed there had he destined it for Jews. Such were the points at issue between the two parties, and the quarrel eventually led to an appeal to arms. (Josephus 1967, II.266–67, p. 427)

He continues as follows:

> Every day, the more venturesome in either camp would rush into combat, for the older members of the Jewish community were incapable of restraining their turbulent partisans, and the Greeks considered it humiliating to give way to the Jews. The latter had the advantage of superior wealth and physical strength, the Greeks that of the support of the military, for the troops stationed here were mainly levied by the Romans from Syria and were consequently always ready to lend aid to their compatriots. The magistrates, indeed, were at pains to repress these disorders and constantly arrested the more pugnacious offenders and punished them with scourge and imprisonment, but the sufferings of those arrested, so far from checking or intimidating the remainder, only served as a stimulus to sedition. (Josephus 1967, II.268–69, pp. 427–29)

In this same revolt, Josephus presents the Syrians at a great disadvantage over the Jews, even going so far as to make theological assessments of that fact as follows:

> The Syrians on their side killed no less a number of Jews; they, too, slaughtered those whom they caught in the towns, not merely now, as before, from hatred, but to forestall the peril which menaced themselves. The whole of Syria was a scene of frightful disorder; every city was divided into two camps, and the safety of one party lay in their anticipating the other. They passed their days in blood, their nights, yet more dreadful, in terror. For, though believing that they had rid themselves of the Jews, each city still had its Judaizers, who aroused suspicion, and while they shrunk from killing off hands of this equivocal element in their midst, they feared these neutrals as much as pronounced aliens. (Josephus 1967, II.461–63, p. 503)

> Thus far, the Jews had been faced with aliens only, but when they invaded Scythopolis, they found their own nation in arms against them, for the Jews in this district ranged themselves on the side of the Scythopolitans, and, regarding their own security as more important than the ties of blood, met their own countrymen in battle. (Josephus 1967, II.466, p. 505)

As can be seen, and in accordance with what has been said above about the way Josephus describes the groups according to their ethnic classification, two groups are identified: the Jews—one side—sometimes identified according to their religious group within Judaism; and the "rest" or the "aliens"—the other side (ἀλλοφύλλων)—regularly identified as "Syrians", but with different classification or denomination according to the region: Greeks, Egyptians, Macedonians, etc. Note what Flavius Josephus says in the following:

> At Alexandria, there had been incessant strife between the native inhabitants and the Jewish settlers since the time when Alexander, having received from the Jews very active support against the Egyptians, granted them, as a reward for their assistance, permission to reside in the city on terms of equality with the Greeks. This privilege was confirmed by his successors, who, moreover, assigned them a quarter of their own in order that, through mixing less with aliens, they might be free to observe their rules more strictly, and they were also permitted to take the little of Macedonians. (Josephus 1967, II.487–88, p. 513)

> But while Ananus was enlisting and marshaling efficient recruits, the Zealots, hearing of the projected attack—for word was brought to them of all people's proceedings—were furious and dashed out the Temple in regiments and smaller units, sparing none who fell in their way. Ananus promptly collected his citizen force, which, though superior in numbers, in arms, and through lack of training, was no match for the Zealots. Ardour, however, supplied either party's deficiencies; those from the city were armed with a fury that was more powerful than weapons, and those from the Temple were recklessness, outweighing all numerical superiority. The former persuaded that the city would be uninhabitable to them unless they were victorious; no form of punishment would be spared them. (Josephus 1968, IV.196–99, pp. 59–61)

> Although these frenzied men had stopped short of no impiety, they nevertheless admitted those who wished to offer sacrifices, native Jews suspiciously and with precaution, strangers after a thorough search. Yet these, though successful at the entrances in deprecating their cruelty, often became casual victims of the sedition, for the missiles from the engines flew over with such force that they reached the altar and the sanctuary, lighting upon priests and sacrificers. Many who had sped from the ends of the earth to gather around this far-famed spot, reverenced by all mankind, fell there themselves before their sacrifices and sprinkled that altar, universally venerated by Greeks and barbarians, with libations of their own blood. The dead bodies of natives and aliens, of priests and laity, were mingled in a mass, and the blood of all manner of corpses formed pools in the courts of God. (Josephus 1968, V.15–17, p. 205)

Categories such as "foreigners" vs. "natives" and "priests" vs. "profanes" are striking, always from the point of view of the Jews as follows:

> The city, now besieged on all sides by these battling conspirators and their rabble, was torn apart by them, its people akin to a massive carcass. Old men and women, in their helplessness, prayed for the coming of the Romans and eagerly looked for the external war to liberate them from their internal miseries. Loyal citizens, for their part, were in dire despondency and alarm, with no opportunity to plan any change of policy and no hope of coming to terms or of flight if they had the will

to do so, for surveillance was maintained everywhere. (Josephus 1968, V.27–30, p. 209)

Here, two new categories confirm our intuition of the categorical opposition of two groups: "the external" vs. "the domestic":

(After the destruction of the temple) The Jewish race, densely interspersed among the native populations of every portion of the world, is particularly numerous in Syria, where intermingling is due to the proximity of the two countries. But it was at Antioch that they especially congregated, partly owing to the greatness of that city but mainly because the successors of King Antiochus had enabled them to live there in security. For, although Antiochus, surnamed Epiphanes, sacked Jerusalem and plundered the temple, his successors on the throne restored to the Jews of Antioch all such votive offerings as were made of brass to be laid up in their synagogue and, moreover, granted them citizen rights on an equality with the Greeks. Continuing to receive similar treatment from later monarchs, the Jewish colony grew in numbers, and their richly designed and costly offerings formed a splendid ornament for the temple. (Josephus 1968, VII.43–45, pp. 517–19)

A certain Antiochus, one of their own number and highly respected for the sake of his father, who was chief magistrate of the Jews in Antioch, entered the theatre during an assembly of the people and denounced his own father and the other Jews, accusing them of a design to burn the whole city to the ground in one night; he also delivered some foreign Jews as accomplices to the plot. On hearing this, the people, in uncontrollable fury, ordered the men who had been delivered up to be instantly consigned to the flames, and all were forthwith burnt to death in the theatre. They then rushed for the Jewish masses, believing the salvation of their native place to be dependent on their prompt chastisement. (Josephus 1968, VII.47–50, p. 519)

According to Rhoads, Josephus reverses this illustration by pointing out that the plague that killed 185,000 of the Assyrian army took place at night when the Jewish hands were at rest or raised in prayer. Thus, he concludes, "Moreover, He knows how, at need, to inflict instant vengeance, as when He broke the Assyrians on the very first night when they encamped hard by; so that had he judged our generation worthy of freedom or the Romans of punishment, He would, as He did the Assyrians, have instantly visited them..." (Josephus 1968, V.407, p. 329). Jewish recourse to arms was therefore futile. In fact, Josephus asserts, the Jewish insistence on fighting and their refusal to respond to repeated Roman offers of surrender were bringing about the destruction of the city and the temple. On the contrary, Josephus describes the Romans doing everything in their power to avoid war and prevent the coming destruction. Josephus thus reverses the Jewish claim of God's eschatological help by arguing that the Jews had not left the matter entirely in God's hands (Rhoads 1976, p. 172). Following the historical account presented by Rhoads (1976, pp. 168–69):

Josephus makes when he says that "foreigners and enemies rectify your (Jewish) impiety" (War VI.102). Josephus attributes to revolutionary ideology the commitment to eliminate Gentile influence in Israel: the high priest rejected Gentile sacrifices and gifts (War II.409); the Sicarii rebuked Jews who cooperated with Rome as "foreigners" and "enemies" (War VII.255; cf. VII.266); and John of Gischala melted down foreign gifts for the temple in order to forge weapons for battle (War V.562). As a reverse polemic to the Jewish suppression of the Gentiles, Josephus stresses the kindness of the Gentiles towards the Jews. Some Jews, he writes, "fled from their countrymen to take refuge with foreigners and obtained from Roman hands the security they despaired of finding among their own" (War IV.397; cf. I.27). Josephus, however, attacks the Jews for mistreating

their own people (War I.27; V.525; VII.266) and for killing Jews "whose lives even the Romans, had they won, would have spared". (War IV.181; cf. VII.266)

Considering that *Jewish War* I–VI was completed in the reign of Titus, during the reign of Domitian, many of Josephus' views and attitudes began to change. It is not entirely clear why. Josephus was becoming more "nationalistic", more conscious of religious considerations, and less concerned with flattering Rome. The usual explanation is that Domitian threatened the status of the Jews, and Josephus came to the defense of his people, or that Domitian suspended Josephus' pension (Cohen 1979, p. 239). Continuing to quote every word in Josephus' works, which evidences a biased "narrative", could be pursued. However, the account presented so far constitutes an overview of a war between some sides and some perspectives: the Jews—with the different denominations explained above—and the non-Jews—also with different ethnic connotations. From a broader understanding, it can even be recognized that in this complex logic of positions and oppositions, of different perspectives and sides, the horizon of valid and socio-religiously accepted human lives versus abject corporealities appears.

## 4. Ethnic Background of the Two Feeding Stories in Mark's Gospel

Considering all that has been elucidated, the insistence of Josephus on designating the adversaries in Galilee as "Syrians" emerges as a noteworthy detail. Additionally, in Mark's narrative of the table conflict (Mk 7), a woman is prominently featured and referred to as "Syro-Phoenician", with an indication of her Greek lineage, which serves as another salient aspect for our analysis. It can be posited that the conflict underlying Mark's Gospel, although manifested as two "incompatible" tables—one for Jews and the other for non-Jews—may indeed have a backdrop shaped by the Jewish war that unfolded between 66–67 CE, culminating in the destruction of the Jewish Temple in 70 CE.

Some commentators who advocate locating Mark's host community in an eastern setting have further argued that Josephus' accounts of the conflicts during the Jewish war explain the mode of the text and the references in the Gospel to persecution, betrayal, and trials (Mk 13:9.13):

Before the 70s, Christians' refusal to get involved in the anti-Roman struggle would have "enraged the revolutionaries", and they would also have been attacked by Gentiles because of their alleged Jewish sympathies, either because they were seen as "Jewish revolutionary groups" or because of their openness towards non-Jews. (Incigneri 2003, p. 82)

Josephus' account shows that such revolts took place mainly in Syria, north of first-century Palestine, which some authors agree is the location of Mark's community. For example, Schenke argues that Mark's community is located "in the Hellenistic cities of the Syrian border region" and was seen as 'a special group of Judaism', which is why Christians were treated in the same way as Jews" (Schenke 2018, p. 23). If the final redaction of this Gospel is placed around the year 70, one will find a direct influence of its aftermath on the new believers, many of whom came from Judaism and many others from the Gentiles. In other words, if Mark's Gospel was composed in the region of Syria, then it can be concluded with a high degree of plausibility that the Jewish War had a direct impact on its recipients and thus on his Gospel narrative (Schenke 2018, p. 143). The discussion about the place of redaction of Mark's Gospel can be followed in Gerd Theissen (1991), *The Gospels in Context: Social and Political History in the Synoptic Tradition*. Theissen situates the discussion in the region of Syria. On the other hand, Joel Marcus (1992), in his paper entitled *The Jewish War*, situates the discussion in the Decapolis (Theissen 1991; Marcus 1992, pp. 441–62; Roskam 2004, pp. 94–113).

Both groups, i.e., Christians coming from Judaism and Christians coming from the Gentiles, directly experienced the "cruel horrors" of the war, but not necessarily at the time of the Neronian persecution, as their experience fits better with the "atmosphere of hatred" towards Jews around the war during 66–69, especially in Antioch (Incigneri 2003, p. 83).

For Marcus (1992, p. 460), Chapter 13, which especially reflects the characteristics of the war, is more easily explained if Mark wrote in geographical and temporal proximity to the events. This author argues that the Gospel was written after the destruction of the Temple in 70, especially because of the precision of Mk 13:1–2, more specifically, "in the shadow of the destruction of the Temple", sometime between 69 and 75.

The preceding analysis highlights elements of the war documented by Josephus, such as attacks on both Jews and Gentiles, Jewish-Gentile tensions, trials of dissenters in Jerusalem, and the imminent threat and eventual destruction of the Temple. This analysis leads to the conclusion that Mark's Gospel alludes to these contemporary events.

Even so, one cannot be absolutely certain of a specific date and place but must work with what can be deduced by putting all these elements together and seeing how they coincide with what is narrated in the Gospel of Mark. Thus, the fundamental question in establishing this background is not about which is the true, unique, and legitimate place where the Gospel was written but in considering the plurality of historical, social, and cultural elements that influenced this theological narrative. While there exists no empirical evidence of Christian persecution in Syria—relying solely on Josephus' testimony—what is apparent is that Gentiles targeted Christians either due to an inability to distinguish them from Jews or simply out of sympathy towards Jews (Incigneri 2003, p. 84). According to Guijarro (2019),

> The trauma to which this statement refers is the social disruption caused by the first Jewish War against Rome, an event that produced a great impact on the land of Israel, on the region of Syria, and on the Judaism of the diaspora. According to this approach, the story of Mark would be an answer to the traumatic experience lived by the followers of Jesus to whom the text was addressed. (p. 141)

This statement advances the understanding of Mark's narrative in terms of the table conflict that is presented since it is no longer just a conflict between Christians of different backgrounds, but, according to this, a conflict between victims and perpetrators, between family members who have experienced the aftermath of the Jewish war at close quarters on both sides. This is supported by statements found in Josephus, such as the followings:

> The Syrians, to be sure, did not make away with a lesser horde of Judeans, but they too would butcher those who were being taken in the cities—not solely out of a hatred that was as before, but now also anticipating the risk to themselves. A terrible disturbance was gripping the whole of Syria, and every city had been divided into two armed camps; safety for one side consisted in anticipating the others. They spent their days in bloodshed, but still more difficult were the nights (they spent) in dread. For they all severally, though thinking it proper that the Judeans had been gotten rid of, continued to hold in suspicion those who were Judaizing, while no one stood quite ready to do away with this ambiguous (element) in the various (places), each feared a mixed (person) as though an actual foreigner. Now, what kept calling forward for the butchering of their foes, even those who had long seemed altogether mild, was greed. For they would pillage with impunity the belongings of those who had been conducted away with and, just as if they were the spoils of those who had been performed away with as a result of battle, they would transfer (the goods) to their own houses. The one who had gained the most was held in honor as having overcome the greater number.
>
> (Josephus 1967, II.461–64, pp. 503–4)

In the Syrian region during and after the war, the events experienced, such as the destruction of the temple and the consequent elimination of worship, forced the Jews to rethink their religious practices. This added to the fact that they came under Roman control, provoked a climate of instability and violence in the region that affected all the inhabitants (Guijarro 2019, p. 149). This situation of anxiety and uncertainty can be inferred from Mark 13 and, if you will, from the somewhat "desperate" plea of the Syrophoenician woman in Chapter 7.

Considering the location of Mark's target community in northern Galilee during the Jewish war and the destruction of the Temple, along with the contributions of Flavius Josephus and the nuances present in his narrative, Mark can be placed in a context of increasing conflict between the Christian and Jewish communities (Horrell 2020). The communities emerged from this moment with a new definition of who was and who was not a Jew. All of this "confrontation" had important effects on Christians, whether they were of Jewish origin, and echoes of some of these conflicts have come down to us through Mark's narrative and elsewhere in the New Testament (Marcus 2004, p. 462).

## 5. Conclusions

Through our investigation, it has been discerned that Mark incorporated two accounts of the feeding of the multitude within his narrative. The discrepancy between these narratives arises from one being traditional (Mk 8:1–10) while the other is a composition unique to Mark (Mk 6:30–44). The section concerning the loaves of bread, spanning Chapters 6–8, warrants particular attention, especially Chapter 7 in its relationship to Chapters 6 and 8. Furthermore, the designation of the woman as a "Syrian" in Chapter 7 is consistent with the clues provided by Flavius Josephus, which suggest that there were internal divisions that fractured relations among the Galilean Jews, especially during the time of the Jewish revolt.

In this segment, it becomes apparent that two distinct groups partake of the bread: those depicted as "like sheep without a shepherd" (Mk 6:34) and those who "have come from afar" (Mk 8:3). However, upon closer examination of the Syrophoenician woman, it becomes evident that her request for the remnants of the bread, as a Greek woman, serves as pivotal evidence regarding the receptivity or lack thereof towards Gentiles within the community to which the Gospel is directed. This evidence underscores the interconnectedness between the salvation offered by Jesus—manifested in the plea of the Syrophoenician woman—and the universal nature of this salvation, extended to both Jews (*sons*) and non-Jews (*puppies*). Thus, Mark's ethnic designation of the woman in Chapter 7 as Greek by birth and Syrophoenician to differentiate her from the (Jewish) "sons" is further evidence that, in the context of Mark's Gospel, the origin of the new believers was an issue, and that Christians coming from Judaism would have some kind of privilege and negative "suspicion" towards Christians coming from the Gentiles.

However, the inclusion of Gentiles did not pose the primary challenge for these communities, as evidenced by Jesus' acceptance and acknowledgment of the faith of a pagan of Phoenician origin (Mk 7:27) and Peter's acceptance of the hospitality of a Gentile (Acts 11:3). The issue they appeared to grapple with pertained to the appropriate treatment of these individuals after their admission. Consequently, it is understandable that Acts 15 addresses the question of whether Gentiles could be admitted to the church without undergoing circumcision and adhering to the law (Acts 15:1.5). The Chapter concludes with the "Apostolic Decree", which does not explicitly address circumcision or obedience to the law. Instead, it focuses on establishing rules and regulations facilitating the coexistence of Jewish and Gentile Christians, including provisions for communal dining, as similarly depicted in Galatians 2:11–14. Peter's hesitance regarding eating with Gentiles, as depicted in Acts, suggests that the Jerusalem church harbored reservations about admitting Gentiles into the church due to the practical implications at the communal table rather than concerns about salvation.

From this problem of openness to the Gentile world, some authors conceive this question. However, in addition to this, it is found that there is a problem of ethnicity and that the Jewish war, which confronted people of both origins, also contributed to the challenge of this openness in the first century AD. The examples given, taken from the historian Flavius Josephus, showed that ethnicity was a problem in which we identified "two sides": the "Jews" and the "non-Jews", with different denominations of geographical origin, mostly designated as "Syrians", the northern border of Palestine where we located the target community of Mark's gospel.

It can be posited that the conflict underlying Mark's Gospel, which is depicted narratively through two "incompatible" tables, one for Jews and one for non-Jews, is rooted in issues of ethnic origin but was further exacerbated by the repercussions of the Jewish war that transpired between 66–67 CE, culminating in the destruction of the Jewish Temple in 70 CE. Although the precise date and location of the composition of Mark's Gospel remain uncertain, an inference drawn from the convergence of various elements suggests a correlation with the narrative presented within Mark's Gospel. Moreover, although there is no concrete evidence of the persecution of Christians in Syria—based only on the testimony of Josephus—it is plausible to claim that the Gentiles attacked the Christians because of their inability to distinguish them from the Jews or because of the closeness they had with the Jews. This is substantiated by Josephus' testimony, which, despite potential nuances in his narrative, describes attacks on both Jews and Gentiles, tensions between the two groups, trials of dissenters in Jerusalem, and the looming threat and eventual destruction of the Temple. Thus, it can be concluded that the Gospel of Mark alludes to these contemporary events.

**Funding:** This research received no external funding.

**Institutional Review Board Statement:** Not applicable.

**Informed Consent Statement:** Not applicable.

**Data Availability Statement:** Data are contained within the article.

**Conflicts of Interest:** The author declares no conflict of interest.

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
