# Peer review of "Ethnic Background of the Two Feeding Stories in Mark’s Gospel"

_religions, doi:10.3390/rel15050553_

Round 1

Reviewer 1 Report

Comments and Suggestions for Authors

Comments on the Quality of English Language

Author Response

Dear Reviewer,

I trust this message find you in excellent spirit.

I write to express my profound gratitude for the thorough evaluation you provided for my work. I was truly impressed by the depth of insight reflected in your comments, evidencing a meticulous examination of my research. Unlike many evaluations I have encountered, yours stood out for its discerning scrutiny, offering invaluable recommendations for enhancing the quality of my study. Each suggestion you provided served as a precise directive for refinement, underscoring your commitment to advancing scholarly discourse.

I wish to extend my heartfelt appreciation for your dedication and the caliber of your feedback, which has served as a catalyst for furthering my research endeavors in a field that resonates deeply with my passions.

I am pleased to inform you that I have meticulously addressed all the points you raised, diligently revising and refining the manuscript in accordance with your guidance, as well as incorporating feedback from other reviewers. The discernible enhancements in the revised version owe much to the invaluable insights you shared.

Lastly, as a person of Christian faith, allow me to convey my gratitude with the expression commonly used in my culture, “God repay you.”

With warm regards,

Reviewer 2 Report

Comments and Suggestions for Authors

See attached file

Comments on the Quality of English Language

See attached file

Author Response

(The authors gave the same response as above.)

Reviewer 3 Report

Comments and Suggestions for Authors

This piece argues that the two feeding stories in Mark (and Matthew, dependent on Mark) make an important point about the inclusion of Jews (Mark 6) and Gentiles (Mark 8), that the whole segment of Mark is carefully structured with the story of the Syro-Phoencian woman in between the two feedings. The reading of Mark is interesting and relies on evidence of ethnic conflict in the region around the time of the Jewish Revolt against Rome to develop a reading of the social situation that Mark's gospel addressed.

The article can be published but it needs some work. Most important will be to attend to the use of texts from Josephus, the Jewish War. There are some general problems with which translation is being used (see the notes below), but more importantly some of the texts cited are not particularly relevant to the social situation in which the paper is interested.

The language of the piece could be made clearer and more succinct.  I have noted a number of particular points for clarification in the detailed comments that follow.

Particular issues

24: What does "According to the above" refer to? The author's previous piece? Why capitalize?

37: The phrase "It is established as a precedent in this research..."  is quite cumbersome. How about just "The following chronology of New Testament texts is presumed: Galatians, etc."  The overall chronology is probably OK, although why Hebrews should be dated after Acts is not clear.

41: It's old fashioned, but introducing a sentence with "But" is not elegant.

46: "interrogation" seems a bit pretentious. How about simply "the question"

49-50: Two forms or instances of "literary analysis" are mentioned. Is that necessary? The whole description of the process of the argument can be simplified.

53: Why exactly is Wheatley being cited here? The whole section is describing what this piece is to do.

93: What does "According to the above" refer to?

110-111: "This puts us on a plane of meaning, narratively speaking, beyond the miraculous action performed by Jesus." It is not at all clear what this means. What does it mean to be "speaking narratively"?

120-121: The final sentence in this paragraph, like the final sentence in the previous paragraph is rather opaque.

130: "is purposeful on Mark's behalf" What does "on Mark's behalf" mean? Do you mean that Mark purposefully sets the story in Tyre to make a point? Nothing is being done "on Mark's behalf."

135: Does Ps 87:4 promise "salvation"? Does it even treat the subject?

136: Jesus': Delete the apostrophe.

138: Firstly: Why not just "First"?

            "deduction". Perhaps "inference" would be better.

143: "According to the internal focus of Jesus." How does one know "the internal focus". And how does that knowledge affect how "one might think."  I think you mean that the movement to pagan territory by Jesus at Mark 7:24 may be attributed to a desire to avoid recognition?

148: Who is the "they"

149-150: The language of "discerned rhetorical intentionality" is rather pretentious. Can the point be more simply stated?

157: What is the antecedent of "this"?

185: "Greek" is to be taken as "an indication of her culture and religion".  Did Greeks have "a religion".

186: "higher social stratum" Why? Is there any indication of social status in the text?

201: What is the subject of "are portrayed"?

208: What is the antecedent of "this"?

244: Breathing and accent on ἔθνος

253: "those who came around the whole of Palestine"? Perhaps "those who came from the whole of Palestine"

255: Some, Delete the comma.

272: Use the nominative and accent: Γαλιλαῖοι.

276-278: The sentence "For Josephus ...place" is gobbledygook. What is - the Sepphoris - doing in the middle of it? Rewrite.

282-283: This sentence seems to suggest that Josephus uses two terms "Judaeans" and "Jews." What might those two terms be? I think you mean that his use of Ἰουδαῖος has two senses.

285: What is "the aforementioned insertion" and what exactly is Frey, et al. being cited for?

287: What is the antecedent of "this"?

296: The date of the Jewish War is here and elsewhere listed as 66-67. Military operations were put on hold after 67 when Vespasian went west to resolve the issue of Nero's successor, but the war went on. The usual dates are 66-70 (or 73 if you want include the mopping up operations after the fall of Jerusalem.)

302: "often" Where outside of Acts are Gentile Christians so called?

313-14: The translation of Heb 13:9 is incorrect. See, e.g., the NRSV: "For it is well for the heart to be strengthened by grace, not by regulations about food, which have not benefitted those who observe them." The key problem is the translation of the verb in the relative clause. It's passive.

321: "place us in a context" Perhaps better: "takes place in a context".

331: It might be better to state the claim and not ask a rhetorical question.

348: Word choice: Is what is involved in Galatians "postponing" ritual matters?

366: The point of the scene in Acts is clear enough, but is the phrase explaining it clear? Perhaps "the common table will be divinely approved."

375: I'm not sure what the "height of resistance" in Jerusalem refers to. Resistance to relaxation of kashrut laws? Why would the introduction of the issue by folk from Antioch have come "at the height of resistance"? Because Stephen was going to be martyred?

381: "delicacies" Really?

439: The reference to Josephus uses some Italian text, not listed in the Bibliography, and what I assume is the Loeb (Harvard 1927). Only one is needed and the normal way of referring to Josephus is by the name of the work, volume and section. So War 2.264-68. The passage cited is actually Jewish War or simply War 2.266. The only problem is that the translation is not that of Thackery in the Loeb. It is an OK translation, but not what is cited.

473: "denomination" Would "classification" or "designation" be better? "Denomination" has religious connotations that don't apply. More importantly, the piece seems to argue that these ethnic designations are based upon region and seems to suggest that "Syrian" is a more generic category. The citation that follows is about Jewish-gentile conflict in Alexandria. Syrians are nowhere to be found. And "Greek" is not a regional "denomination" of "Syrian." The translation that is used here introduces the word "Gentile" (line 482), which may introduce an inappropriate technical term. The Greek is ἀλλοφύλλων, which Thackeray appropriately translates as "aliens."

485-494: This passage comes from War 4.198-200 which describes the situation in Jerusalem early in the revolt. The passage says nothing about the topic of  hostility between Jews and gentiles (esp. Syrians). This passage in Josephus is all about internal divisions among Jews. The enemy, the Zealots, are internal not external.

496-503: This passage is from War 5.15-19, more precisely from 5.17. This passage describes the brutal chaos within the Temple precincts in which the various Jewish factions slew one another. It has nothing to do with the issue on the table.

586: What is the reference to "such revolts"?

596-598: Theissen's book was originally published in German: Lokalkolorit und Zeitgeschichte in den Evangelien : ein Beitrag zur Geschichte der synoptischen Tradition (Göttingen : Vandenhoeck & Ruprecht, 1989). Why cite the Spanish translation?

604-05: It is not clear what the reference to the Neronian persecution is supposed to be.

611: The "precision" of Mark 13:1-2 might be specified, "no stone on another."

620-62:1 It is difficult to see see how the two part of this sentence hang together. "While there is no empirical evidence of Christian persecution in Syria" clashes with "what is apparent is that Gentiles targeted Christians..." What is the basis for the second clause? How can something for which there is no evidence be apparent? The claim is repeated in conclusion (lines 721-22).

623: What is the antecedent of "this statement" in the first line of the citation?

634-650: This paragraph from Josephus, War 2.461-64 offers a vivid description of the chaos in Syria at the early stage of the revolt and the way in which it affected Jews. What introduces the paragraph is a note about the "aftermath of the Jewish war." It would be nice if the cited material actually spoke about the aftermath.  One can infer that the situation between "Syrians" and "Jews" (including Christians?) was difficult after the war, but the text is dealing what led to that the presumed post-war situation.

653: The destruction of the Temple in Jerusalem caused more than in "interruption" of worship. "Elimination" perhaps?

Comments on the Quality of English Language

Included above.

Author Response

(The authors gave the same response as above.)

Reviewer 4 Report

Comments and Suggestions for Authors

This is a fine piece of work. At most I would have expected a little more attention to the two feeding accounts underlining Mark's delineation of them both by location and symbolism (12 7 5000 4000 etc). Perhaps also reference to the similar juxtaposition of Gentile and Jewish (12s) in Mark 5.

Author Response

(The authors gave the same response as above.)

Round 2

Reviewer 3 Report

Comments and Suggestions for Authors

I noted that there remains a significant problem with the citation of passages from Josephus. The passages cited in lines 515-37 do not illustrate hostility between Jews and Gentiles. They depict inner Jewish divisions in the city of Jerusalem during the Jewish revolt.

The citation style for Josephus also needs to be made uniform.

Comments on the Quality of English Language

See my initial comments on this piece.

Author Response

Dear Reviewer,

I trust this correspondence finds you in excellent spirits. 

I wish to express my sincere appreciation for the meticulous review you have conducted on my manuscript. Each of your insights has proven invaluable, enriching my understanding of the subject matter. Comparing the initial draft with the refined version underscores the notion that continuous improvement is not only possible but essential in scholarly endeavors.

It is my earnest hope that all your recommendations will be embraced with the same enthusiasm with which they were imparted to me.

With utmost gratitude and warm regards,